# Feasibility Study on the Use of Recycled Polymers for Malathion Adsorption: Isotherms and Kinetic Modeling

**DOI:** 10.3390/ma13081824

**Published:** 2020-04-12

**Authors:** Jhonatan J. Hermosillo-Nevárez, Victoria Bustos-Terrones, Yaneth A. Bustos-Terrones, Perla Marysol Uriarte-Aceves, Jesus Gabriel Rangel-Peraza

**Affiliations:** 1Tecnológico Nacional de México - Instituto Tecnológico de Culiacán, Juan de Dios Bátiz 310, Col. Guadalupe, Culiacán 80220, Mexico; 2Departamento de Ingeniería en Tecnología Ambiental, Universidad Politécnica del Estado de Morelos, Boulevard Cuauhnáhuac 566, Col. Lomas del Texcal, Jiutepec 62550, Mexico; 3CONACYT - Instituto Tecnológico de Culiacán, Juan de Dios Bátiz 310, Col. Guadalupe, Culiacán 80220, Mexico

**Keywords:** polyvinylchloride, polystyrene, adsorption, malathion, isotherms

## Abstract

In this study, the use of Polyvinylchloride (PVC) and High Density Polystyrene (HDPS) was demonstrated as an alternative for the adsorption of Malathion. Adsorption kinetics and isotherms were used to compare three different adsorbent materials: PVC, HDPS, and activated carbon. The adsorption capacity of PVC was three times higher than activated carbon, and a theoretical value of 96.15 mg of Malathion could be adsorbed when using only 1 g of PVC. A pseudo first-order rate constant of 1.98 (1/h) was achieved according to Lagergren kinetic model. The adsorption rate and capacity values obtained in the present study are very promising since with very little adsorbent material it is possible to obtain high removal efficiencies. Phosphorous and sulfur elements were identified through Energy Dispersive X-ray (EDX) analysis and evidenced the malathion adsorption on PVC. The characteristic spectrum of malathion was identified by the Fourier Transform Infrared (FTIR) Spectroscopy analysis. The Thermogravimetric and Differential Thermal Analysis (TG/DTA) suggested that the adsorption of malathion on the surface of the polymers was mainly determined by hydrogen bonds.

## 1. Introduction

In recent years, the adsorption process has acquired great technological relevance due to its implementation for the removal of pollutants in the environment. A contaminant can be separated from a gas or liquid phase due to a physical phenomenon that can occur by the Van der Waals forces and hydrophobic interaction (not covalent bonding) on the surface of the adsorbent material [1,2,3,4].

The adsorption process has been widely used for the removal of a great diversity of contaminants in water, such as drugs, pesticides, heavy metals, dyes, among others [5,6,7,8,9,10]. Ji et al. [11] report the use of nonporous graphite, single walled carbon nanotubes and commercial microporous activated carbon for the removal of three antibiotics: sulfamethoxazole, tetracycline, and tylosin. Demirbas et al. [12] used activated carbon derived from hazelnut shells for the adsorption of Cu (II) ions present in aqueous solution. Benjwal et al. [13] carried out the adsorption of methylene blue on feather fiber-derived carbon fibers.

In addition, the adsorption process has also proven to be a viable alternative for the removal of pesticides. Vukčević et al. [2] report the use of activated carbon from waste hemp (*Cannabis sativa*) fibers for the removal of pesticides, such as acetamiprid, dimethoate, nicosulfuron, carbofuran, and atrazine. Taha et al. [14] used carbon obtained from the biomass pyrolysis (biochar) to remove a mix of 15 pesticides (azinphos-methyl, phosmet, boscalid, chlorfenvinphos, fluotanil, diazinon, carbaryl, malathion, imidacloprid, acetamiprid, propiconazole, flusilazole, triadimenol, atrazine, and oxamyl).

In the present study, the removal of malathion is of great interest because it is one of the most used organophosphate pesticides. This pesticide is commonly found in agricultural effluents and waterbodies in regions with intense agricultural activity [15,16]. In this sense, several efforts have been reported for the removal of malathion using the adsorption process. Jusoh et al. [17] evaluated the performance of commercial granular activated carbon prepared from coconut shells and palm for the removal of malathion. Habila et al. [18] report the malathion adsorption by activated carbon derived from a copyrolysis process of agricultural and municipal solid wastes, such as paper, plants, and plastics. Mirković et al. [19] report the use of mesoporous monetite as efficient adsorbent of malathion from aqueous solutions. Harper Jr. et al. [20] demonstrated the feasibility of use of metallic materials such as iron and copper for malathion adsorption, but the adsorption capacity was low compared to the typical adsorbent materials used (activated carbon). Wanjeri et al. [21] went a step further and suggested the use of adsorbent materials with nanoparticles: graphene oxide coated with silica nanomagnetic particles was used for the removal of chlorpyrifos, parathion, and malathion.

However, the preparation of the adsorbents used in the studies aforementioned implies high costs due to the use of thermal, chemical, or mechanical processes. This investigation proposes a low cost alternative for malathion adsorption by reusing polymer waste materials that do not require prior treatment for their application. In this sense, this paper describes the malathion adsorption on two polymeric materials: PVC and HDPS. Then, their adsorption performance were compared with the reference adsorbent material: activated carbon. The objective of this study is to evaluate the feasibility of using these polymeric materials in the malathion adsorption process. The adsorption kinetics and isotherms were evaluated and a comparative statistical analysis of the adsorption process efficiency is given.

## 2. Materials and Methods

### 2.1. Adsorbate

The adsorbate used was Malathion, Diethyl 2-[(dimethoxyphosphorothioyl) sulfanyl] succinate, obtained from the local market (Velsimex e-1000). This pesticide had a technical degree with purity of 87.8%. This molecule is characterized by a thiol group attached to the phosphate through a double bond, which is an indicator of the high persistence of the pesticide.

### 2.2. Preparation of Adsorbent Materials

#### 2.2.1. Activated Carbon

The activated carbon was prepared through a process of thermal activation. The carbonization process was carried out at 700 °C under purified nitrogen (99.995%) for 2 h. Potassium hydroxide (KOH) solution was used to soak the carbon in a proportion of 1:3 (Char:KOH, wt.%). The mixture was then dehydrated in an oven overnight at 105 °C. Then, the carbon was activated to a final temperature of 800 °C. When temperature was achieved, the nitrogen gas flow was switched to carbon dioxide and these conditions were held for 2 h. The activated carbon was then cooled to room temperature. Hydrochloric acid (0.1 M) was used to eliminate impurities and then activated carbon was washed with deionized water until the pH of the washing solution reached 6.5–7 [22].

#### 2.2.2. Polyvinylchloride

Recycled foamed PVC was used as adsorbent material in this study. This polymer was washed with distilled water and put in an oven at 105 °C to eliminate excess water. Small particles of 0.25 cm by 0.25 cm were shaped by hand. The resultant cut particles were used in adsorption experiments. No chemical treatment was needed for the activation of this material.

#### 2.2.3. Polystyrene

The HDPS used in adsorption experiments was also a recycled material. The same procedure was used for its preparation: this material was washed and the excess of water was eliminated by thermal treatment (105 °C). Then, small particles of 0.25 cm by 0.25 cm were created. No chemical treatment was needed for the activation of this material.

### 2.3. Experimental Design

The optimization of the adsorption process was carried out by using a Taguchi L9 orthogonal array. The approach of Taguchi L9 is used to achieve optimal operating conditions through optimization functions. This approach is used in many areas of knowledge whenever the settings of interest parameters are necessary. Taguchi proposes the use of a set of arrays called orthogonal arrays, which provide the way of conducting the minimal number of experiments and provide the full information of all the control variables that affect the performance parameter [23]. In this study, the temperature, the type of adsorbent material and the volume occupied by the dry adsorbent material were used as control variables in this study (Table 1). The operation levels for each control variable (treatments) are shown in Table 2. The order of the treatments carried out in this study was randomized. Synthetic wastewater with a Malathion concentration of 51.72 mg/L was prepared with distilled water. A pH value of 4.5 was obtained after synthetic wastewater was prepared. The treatments took place in a batch reactor of 1 L with agitation and control of temperature. The different adsorbent materials were added in this reactor according to the conditions established in the L9 orthogonal array. An analysis of variance (ANOVA) was carried out to figure out the best operating conditions for the adsorption process.

### 2.4. Quantification of the Adsorbate

As it was mentioned before, the adsorbate used in this work was Malathion. Malathion was extracted based on the AOAC 970.52 methodology. The extraction procedure of Malathion was carried out as follows: 200 mL of methylene chloride were used to extract Malathion from 1 L of synthetic wastewater. The sample was placed in a separating funnel and mixed during three minutes for the separation of phases. The extract obtained was placed in a Kuderna Danish flask with heating to obtain 1 mL of concentrated sample. This extraction procedure showed pesticide recovery rates ranging from 81% to 96%, with standard deviations less than 5%.

Gas Chromatograph—Electron Capture Detector (GC-ECD Perkin Elmer model Clarus 680) and Mass Spectrometry (Perkin Elmer model Clarus SQ 8C) were used for the determination of malathion. Stock standard solutions of malathion (97.5%) purchased from Fluka (Sigma-Aldrich Laboratories, St. Louis, MO, USA) were prepared in pesticide-grade hexane (Baker, Pittsburgh, PA, USA) to the concentration of 100 mg/L and stored at 4 °C. Stock solutions were prepared to obtain a calibration curve. By using the calibration curve, the concentration of Malathion was measured in this study.

### 2.5. Adsorption Kinetics

The adsorption kinetics was investigated by using the Lagergren pseudo-first order model. The Lagergren pseudo-first-order kinetic equation in its linear form is [22]:(1)$$log\left(q_{e}-q_{t}\right)=log\left(q_{e}\right)-\left(\frac{k_{1}}{2.303}\right)t$$
where $q_{t}$ is the amount of malathion adsorbed (mg/g) at time t (h) and $k_{1}$ is the pseudo first-order rate constant (1/h).

### 2.6. Equilibrium Studies (Isotherms)

The adsorption isotherm was obtained and the performance of the adsorbent materials was evaluated. Seven experiments were carried out for the construction of adsorption isotherms, where the quantity of material mass was varied according to Table 1. These experiments were independent from the ones carried out for adsorption kinetics. The calculations of the isotherms were performed using Langmuir and Freundlich linear models. The linear model of the Langmuir isotherm is the following [22]:(2)$$\frac{c_{e}}{q_{e}}=\frac{1}{Q_{0}}+\frac{1}{bQ_{0}}\left(C_{e}\right)$$
where $q_{e}$ is the quantity of Malathion adsorbed in a given quantity of adsorbent (mg/g), $Q_{0}$ and *b* are constants of Langmuir and $C_{e}$ is the concentration in balance in mg/L. The adsorbed quantity of Malathion at equilibrium ($q_{e}$) was calculated by:(3)$$q_{e}=\frac{(C_{0}-C_{e})V}{W}$$
where $C_{0}$ and $C_{e}$ (mg/L) are the liquid-phase concentrations of malathion at initial and equilibrium, respectively. V (L) is the volume of the solution and W (g) is the mass of dry adsorbent used. The model of the Freundlich isotherm is the following:(4)$$Lnq_{e}=LnK_{f}+\frac{1}{n}LnC_{e}$$
where 1/*n* is the constant that represents the constant of saturation of the adsorbate and $K_{F}$ is the empirical constant that indicates the capacity of adsorption and the affinity of the adsorbate for the adsorbent.

### 2.7. SEM/EDX Analysis

The morphology and elemental analysis of the adsorbent were characterized with SEM (SU1510, Hitachi, Tokyo, Japan) – EDX (Genesis, Edax, Stoke-on-Trent, UK) equipment. Scanning Electron Microscopy (SEM) was employed to study the morphology of the material, while Energy dispersive X-ray (EDX) was used to examine the elements on the surface of the adsorbent. The elemental analysis in EDX mode included: carbon, oxygen, aluminum, phosphorous, copper, and sulfur.

### 2.8. Fourier Transform Infrared (FTIR) Spectroscopy Analysis

FTIR analysis was applied to determine the spectrophotometric trace of malathion present on the adsorbent. This analysis was made using a PerkinElmer Paragon 1000 infrared spectroscope with Attenuated total reflectance (ATR) at a wavelength range of 400–4000 (1/cm) with 24 scans. An infrared beam was put through the material samples with the objective of verifying qualitatively the adsorption of malathion on the polymeric material; the analysis was done before and after the adsorption tests. Through Fourier transform, the results of the spectroscopy turn into a spectrum with the structural information of the different molecules [24].

### 2.9. Thermogravimetric and Differential Thermal Analysis (TG/DTA)

Thermogravimetric and differential thermal analysis (TG/DTA) is commonly used for studying polymeric materials. In this work, this analysis was carried out to describe the behavior of the physical properties of the adsorbent, as well as determining the possible presence of malathion adsorbed on the material. TG/DTA analysis was carried out with a PerkinElmer Diamond Thermogravimetric and Differential Thermal Analyzer. A temperature gradient of 10 °C/min was used in an interval of 50 °C to 600 °C under an inert atmosphere. TG/DTA analysis was carried out in the polymeric samples before and after the adsorption of malathion.

## 3. Results and Discussion

### 3.1. Statistical Analysis

Table 2 shows the results of the Taguchi L9 design applied for the adsorption of Malathion. As it is shown in this table, a great variation of the removed concentration of Malathion is observed in the treatments performed in this study. The Malathion concentration removal varied from 6.5 mg/L to 40.48 mg/L between treatments. The best removal efficiency was found in treatment 5.

The best analysis of variance (ANOVA) is presented in Table 3. The results showed that the amount of adsorbent material is critical in the adsorption process: the greater the amount of adsorbent material is used, the greater the removal efficiency is observed. The best ANOVA also demonstrated that the type of adsorbent material used had a significant effect on the efficiency of the adsorption process. This statistical analysis demonstrated that both polymers showed a greater adsorption capacity than the reference material (activated carbon). This situation demonstrated the hypothesis of this work: the polymer materials without any previous chemical treatment could be used as adsorbents. The reference material for the adsorption process showed a mean removal efficiency of 50%, while HDPS and PVC polymers showed mean removal efficiencies of 72.8% and 73.1%, respectively. These values were significantly higher than the efficiency obtained with reference material (p<0.05).

Based on ANOVA results, temperature showed no effect on the adsorption of Malathion. Other studies reported that adsorption rates increase with increasing temperature [25,26]. Most of the adsorption studies reported an effect of temperature on adsorption process because lower values of temperature were used for this comparison [25]. In this study, the levels used for temperature were 20 °C, 30 °C, and 40 °C. Since these adsorbent materials are considered to be used as packed column for the adsorption of this pesticide in situ in agricultural crops; therefore, these temperature values better describe the behavior of the adsorbent materials in tropical regions, where extensive agriculture activities are carried out.

### 3.2. Adsorption Kinetics

Figure 1 provides important information about the behavior of the adsorption process for each treatment. These results describe the adsorption rate, as well as the time needed for the saturation of the adsorbent material. Figure 1a shows that the activated carbon, PVC, and HDPS achieved a maximum removal efficiency of 96% when using the maximum volume of dry adsorbent material. Under these conditions, activated carbon was saturated after 60 min, while PVC and HDPS were saturated after 80 and 90 min, respectively.

Malathion adsorption on powdered activated carbon is reported by Kumar et al. [27]. In this study, an initial malathion concentration of 50 mg/L was used and a removal efficiency of 94.87% was achieved after 150 min. The time required for the removal of malathion can be considered long in comparison with PVC and HDPS. Furthermore, Hammed et al. [22] report that 240 min were needed for the adsorption of the 2,4-D pesticide on activated carbon derived from date stones. The time required for the removal of this pesticide can be considered high, but this situation can be related to the initial concentration used for this pesticide (400 mg/L). Both aforementioned adsorption studies were carried out at room temperature (30 °C) and under acidic conditions (pH values below 4).

Figure 1b shows the experimental data adjusted to Lagergren pseudo first order kinetic model in its linear form (Equation (Equation 1)). The slope of the linear equation obtained from experimental data represents the pseudo first-order rate constant (*k*). Based on the determination coefficient ($r^{2}$), the PVC and activated carbon adsorption processes fitted well with the theoretical values ($r^{2}$ > 0.95), while HDPS adsorbent material (treatment 5) did not meet the linear behavior of the pseudo first order kinetic model ($r^{2}$ < 0.90). In this study, the pseudo first-order rate constant (*k*) varied from 0.21 to 2.994 1/h. In particular, treatments 7 and 8 showed the highest adsorption rates, with values of 2.994 and 2.082 1/h. These treatments correspond to activated carbon and PVC adsorption processes, respectively. These values were higher than the reported by Hameed et al. [22], who found a value of k=1.29 1/h for the adsorption of 2,4-D pesticide.

### 3.3. Adsorption Isotherms

The behavior observed in the adsorption process when polymers were used for the removal of Malathion (treatments 5 and 8) is an indicator that these adsorbent materials can represent a good adsorbent material alternative. In particular, this situation was analyzed for PVC and HDPS using the adsorption isotherms. Seven experiments were performed using activated carbon, PVC, and HDPS as adsorbent materials, according to Table 1. Three repetitions were carried out for each experiment, where the mass of each of the adsorbent material was varied to better understand the behavior of the adsorption process. Figure 2 shows the results obtained from these experiments based on Langmuir (Figure 2a) and Freundlich (Figure 2b) isotherms’ models.

Based on the values obtained for the determination coefficient ($r^{2}$), the Langmuir isotherm model was the most suitable model to represent the adsorption process in this study. Figure 2a shows the Langmuir isotherms for activated carbon, PVC, and HDPS adsorption processes. Langmuir isotherm for activated carbon shows that, if a greater mass of Malathion is desired to be adsorbed, a large amount of adsorbent material mass is required. In contrast, the Langmuir isotherm for both polymers shows a slight variation of the adsorbent material mass for the removal of large amounts of this pesticide.

In this study, 7.5 g of activated carbon was needed for the adsorption of 41.16 mg of Malathion. According to Langmuir isotherm for PVC, only 1 g of the adsorbent material can adsorb around 96.15 mg of Malathion. Therefore, the adsorption capacity of the material PVC is higher than activated carbon. The values obtained in the present study are very promising since with very little adsorbent material it is possible to obtain high removal efficiencies. The adsorption experiments using HDPS did not fit well ($r^{2}$ < 0.90) with the Langmuir and Freundlich linear models. This result coincides with the obtained in adsorption kinetics, where HDPS experiments did also not meet the pseudo first-order adsorption kinetics.

Langmuir isotherm is also expressed in terms of a dimensionless constant separation factor $R_{L}$ that is given by Equation (Equation 5) [22]:(5)$$R_{L}=\frac{1}{1+bC_{0}}$$

In Equation (Equation 5), the value of $R_{L}$ indicates the shape of the isotherm. The adsorption process could be unfavorable ($R_{L}$ > 1), linear ($R_{L}$ = 1), favorable (0 ≤$R_{L}$ < 1), or irreversible ($R_{L}$ = 0). In this study, the $R_{L}$ obtained for PVC was 0.944, while $R_{L}$ obtained for activated carbon was 0.073. These results indicate that the adsorption of Malathion on foamed PVC is favorable and almost linear. This situation may be related to the characteristics of PVC: the material has a homogeneous adsorption surface with a great superficial area. The experimental results also show that the saturation of the material was carried out as a monolayer of solute molecules on the adsorbent surface [28].

Table 4 shows the results of the adsorption process of Malathion in other studies. Based on this comparison, PVC can be considered a good adsorbent material for this pesticide. The $R_{L}$ coefficient shows that the behavior of the adsorbent material against the adsorbate is favorable, according to the classification shown in [22]. The value of $Q_{0}$ indicates the maximum theoretic concentration that the material can adsorb to form a monolayer and saturate this material completely. The value obtained in this study is superior to that reported in other studies. $Q_{0}$ values of 4.29 mg/g, 16.13 mg/g and 21.74 mg/g are reported in [18] when using activated carbon made from rice husk, while a $Q_{0}$ of 32.11 mg/g is reported in [27] when commercial activated carbon was used. In this study, a $Q_{0}$ value of 96.15 mg/g was obtained, which is three times the highest $Q_{0}$ value reported in literature. Coefficient b obtained from Langmuir model refers to the affinity of adsorption sites of the PVC material by the substrate. According to the Langmuir model, the smaller this coefficient is, the greater the affinity of the adsorbent material for the substrate is [29]. In this study, values of 0.001 L/mg and 0.343 L/mg were found for PVC and activated carbon, respectively. These results demonstrated that PVC has a higher affinity than activated carbon and other adsorbent materials, such as the suggested in Kumar et al. [27] who report values of 0.17 L/mg and 0.53 L/mg.

### 3.4. SEM/EDX Analysis

Once it was demonstrated that foamed PVC showed the highest adsorption capacity, the morphological structure, superficial area and elemental analysis of this adsorbent was characterized (Figure 3). This image was obtained through scanning the surface of this polymer with an amplification of 100X before and after the adsorption process. According to Figure 3b, the material presents a morphology that belongs to crosslinked polymers, according to the classification presented by Fu et al. [30]. In Figure 3a, the elemental analysis of PVC is shown. This analysis corresponds to the adsorbent material that was analyzed before the adsorption process. The polymer of interest is composed of carbon, oxygen, and aluminum. The elemental analysis of this material after the adsorption process (Figure 3c) demonstrated the presence of new elements: phosphorus and sulfur. According to the chemical composition of Malathion, these elements are present in its molecules. Table 5 shows the values of the identified elements by the EDX analysis before and after the adsorption process. This table corroborated quantitatively the presence of these elements on the adsorbent material.

Figure 3b,d show the surface image of PVC obtained through the SEM analysis before and after the adsorption process. When using SEM, it was demonstrated that most of organic compounds are deposited on the inner surface of the PVC fibers. This situation coincides with the mentioned in previous studies, where the bulky macromolecular shape and neutral character of pesticides make organic compounds prone to be adsorbed on inorganic microfibers [29]. Malathion adsorbed on the surface of the polymer is appreciated with an intense white color in Figure 3d. It is also notable that the morphology of the foamed PVC material facilitated the pesticide to become adsorbed.

### 3.5. Fourier Transform Infrared Spectroscopy (FTIR)

In this study, FTIR was used for verifying qualitatively the adsorption of malathion into the polymeric arrays. This was done through IR characterization of the adsorbent material before and after the adsorption process (Figure 4). Figure 4a presents the spectrum of IR transmittance of the polymeric material before adsorption process. A band between 500 (1/cm) and 694 (1/cm) is observed which is assigned to the stretching vibrations of the C−Cl bond. The absorption bands at 752 (1/cm) correspond to the vibrations of the C−H bonds. In addition, adsorption bands were identified in the interval between 1435 (1/cm)–1599 (1/cm) which correspond to the vibrations of an angular deformation of the bounds $CH_{2}-Cl$. Other adsorption bands that can be observed in the spectrum correspond to the materials used as additives in the fabrication of the PVC material.

Figure 4b corresponds to PVC material after adsorption process. In this figure, the bands around 3200 (1/cm)–3400 (1/cm) indicate the presence of water, given that these wavelengths are attributed to the stretching vibrations of =OH bounds. The presence of water is due to the synthetic water used for adsorption experiments. The IR spectrum of the PVC after the adsorption of malathion shows the characteristic behavior of the functional groups that are present in the adsorbed molecule. The last adsorption bands on the far left of Figure 4b at 789 (1/cm) belong to the bounds =CH of the benzene ring, while 690 (1/cm), 650 (1/cm) y 534 (1/cm) bands are part of the characteristic spectrum of malathion. Other adsorption bands that appear can be assigned to the functional groups of materials used to stabilize malathion.

### 3.6. Thermogravimetric and Differential Thermal Analysis (TG/DTA)

Thermogravimetric (TG) analysis is the most widely used technique for the study of polymers, since it allows for describing the behavior of the decomposition of these compounds. The thermal decomposition of polymers can be described due to TG analysis is carried out under an inert atmosphere. This feature was used to identify the presence of Malathion on the adsorbent material. The curves of the TG analysis of PVC before and after the adsorption of malathion are presented in Figure 5. In this figure, the red line shows the endothermic (evaporation and melting) and exothermic processes (crystallization and degradation) that the sample undergoes during the thermal treatment. The blue line shows the weight variation of the sample throughout the TG analysis. In Figure 5a, three typical regions of the degradation of polymers are distinguishable: the first region is characterized by a thermal stability of the polymer material and only 0.4% of its initial weight is lost. This behavior is due to the release of strongly adsorbed water, volatilization of residual solvents and of monomers that are present within the array [31].

Approximately 98.3% of PVC is degraded in the temperature interval of 250–445 °C. Finally, a total degradation of the polymer occurs close to 600 °C. The thermogram in Figure 5b shows a weight loss of 65.6% when a temperature achieved 105 °C, due to the presence of weakly water adsorbed in the matrix. This decomposition process is totally endothermic, as it is demonstrated in the heat flow graph (red line). Between 105–180 °C, approximately 11.6% of weight loss is observed, which could be caused by the decomposition of the water absorbed and the presence of volatile organic compounds and the decomposition of malathion. Finally, the weight loss registered in the temperature interval of 180–600 °C could be related to the degradation of PVC.

### 3.7. Forthcoming Investigations

Based on the results obtained in this study, the technical feasibility for the use of foam PVC as adsorbent filter for the removal of malathion was demonstrated. However, in order to use this material for the removal of pesticides in situ in agricultural crops, in depth research is needed to develop a device that allows the removal and replacing of the adsorbent material once it reaches its saturation capacity. An advanced oxidation process, such as photocatalysis or Fenton process, could be used for the mineralization of the pesticide loaded on adsorbent material. The mineralization of pesticides after adsorption process could demonstrate the technical feasibility of reusing this adsorbent material. In addition, the scale-up of an adsorption device for the removal of pesticides in agricultural crops under a continuous flow is another research area that must be explored.

## 4. Conclusions

This study showed the feasibility of use of two polymers (PVC and HDPS) for the removal of malathion in synthetic wastewater through a physical process of adsorption. The removal efficiency of this process was compared to the reference material: activated carbon. The adsorption efficiency was significantly higher (p<0.05) when using these polymers. In addition, a favorable behavior of the adsorption process was found when these materials were used. This behavior is strongly recommendable in adsorption processes and just a few materials have been found in literature with this characteristic. According to the equilibrium Langmuir model, a theoretical value of 96.15 mg of malathion could be adsorbed when using only one gram of PVC as adsorbent material. This value is almost three times the adsorption capacity of activated carbon. In addition, the first order rate constant obtained for the adsorption of malathion on PVC (2.082 1/h) was similar than the treatment carried out on activated carbon (2.994 1/h). SEM/EDX analysis and FTIR Spectroscopy could demonstrate the adsorption of malathion on these polymers. Based on the results obtained in this study, polymer materials could represent an inexpensive and viable alternative for the removal of pesticides in situ in agricultural effluents. The use of recycled plastic materials as adsorbents could reduce the environmental impact of these materials in diverse ecosystems and could contribute with the reduction of pesticides in water bodies due to agricultural activity.

## Figures and Tables

**Figure 1 materials-13-01824-f001:**
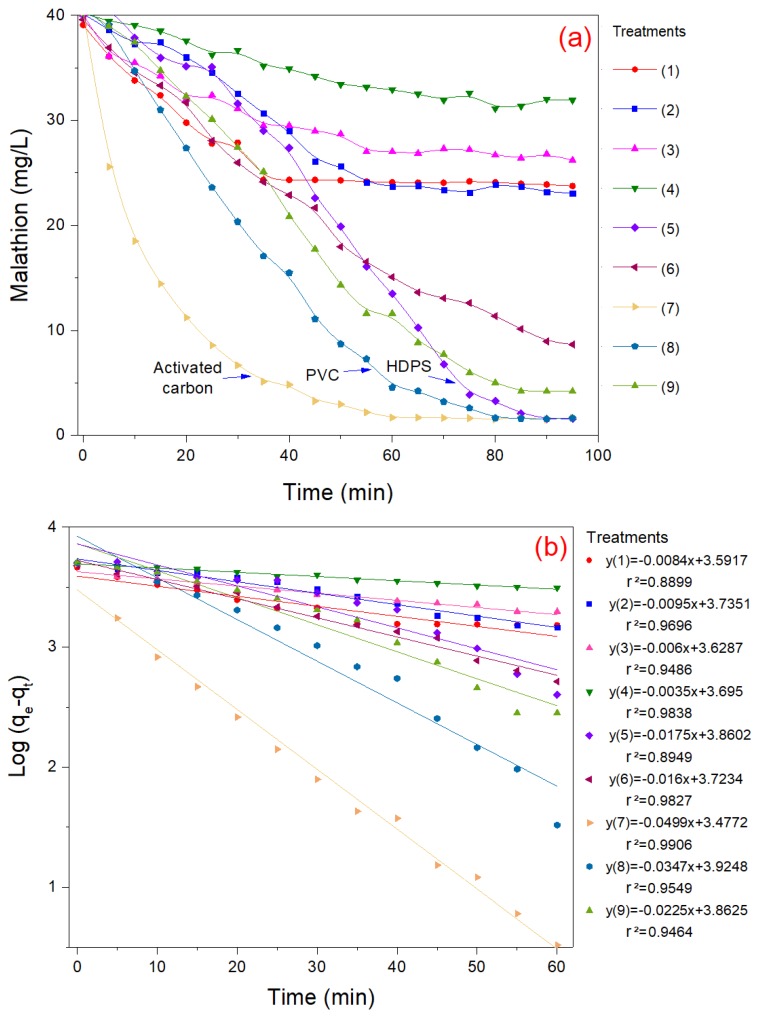
Behavior of malation adsorption process under different conditions (**a**) and its kinetic modeling (**b**).

**Figure 2 materials-13-01824-f002:**
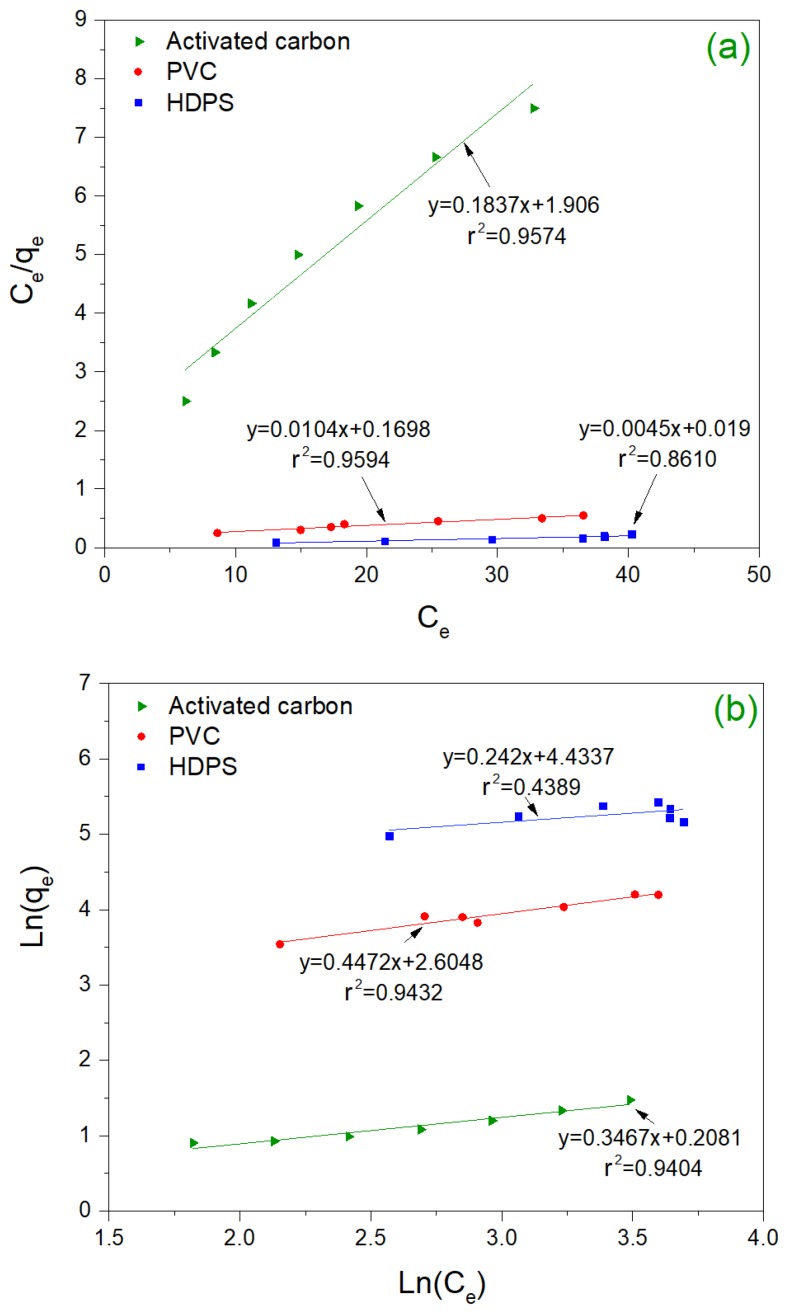
Results of Langmuir (**a**) and Freundlich (**b**) isotherms models for malathion adsorption on PVC and activated carbon.

**Figure 3 materials-13-01824-f003:**
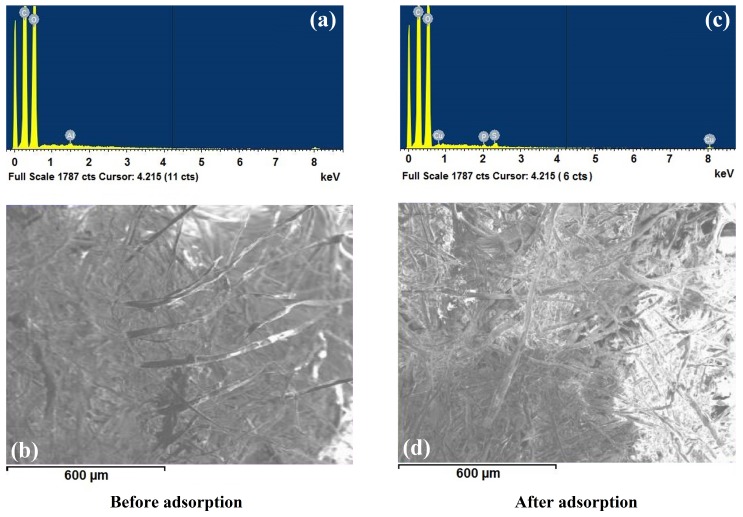
Elemental analysis (**a**,**c**) and SEM images (**b**,**d**) of PVC before and after the adsorption process.

**Figure 4 materials-13-01824-f004:**
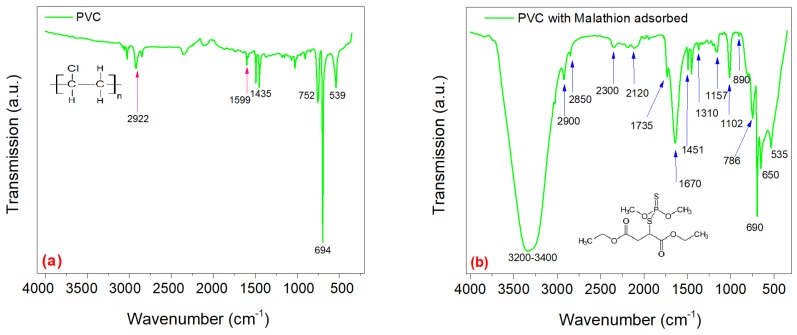
FTIR analysis of PVC before (**a**) and after (**b**) the adsorption process.

**Figure 5 materials-13-01824-f005:**
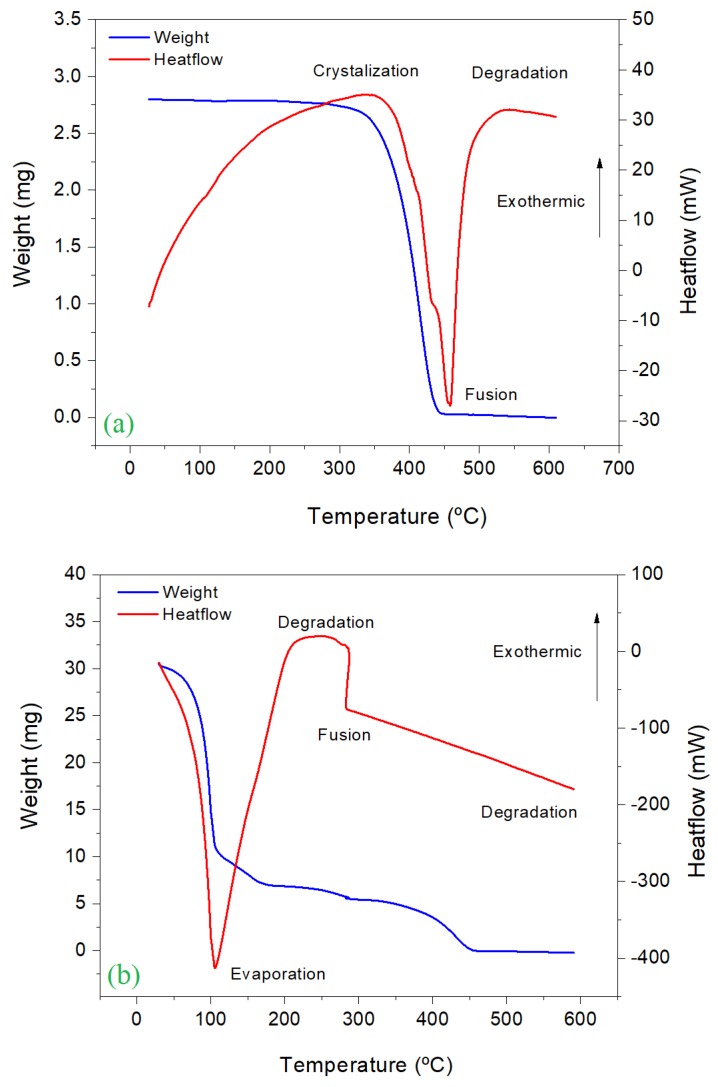
Thermogravimetric analysis of PVC before (**a**) and after (**b**) the adsorption process.

**Table 1 materials-13-01824-t001:** Masses of adsorbents used for the adsorption isotherms experiments.

Volume Occupied by the Dry Adsorbent Materials (cm^3^)	Mass of Dry Activated Carbon (g)	Mass of Dry PVC (g)	Mass of Dry HDPS (g)
16.95	7.50	0.55	0.23
15.07	6.66	0.50	0.20
13.18	5.83	0.45	0.18
11.30	5.00	0.40	0.16
9.42	4.16	0.35	0.13
7.53	3.33	0.30	0.11
5.65	2.50	0.25	0.09

**Table 2 materials-13-01824-t002:** Results of the experimental runs carried out based on L9 orthogonal array.

Treatment	Adsorbent Material	Temperature (°C)	Volume of the Dry Adsorbent Material (cm^3^)	Concentration Removed (mg/L)	Efficiency
	Factor A	Factor B	Factor C		
1	AC	30	11.30	14.74	38%
2	PVC	30	5.65	16.08	40%
3	HDPS	40	5.65	13.06	33%
4	AC	20	5.65	6.17	18%
5	PVC	20	16.95	40.48	96%
6	PVC	40	11.30	32.92	83%
7	AC	40	16.95	38.74	96%
8	HDPS	30	16.95	38.87	96%
9	HDPS	20	11.30	36.50	90%

**Table 3 materials-13-01824-t003:** Best ANOVA for the L9 orthogonal array.

Source of Variation	Sum of Squares	Degrees of Freedom	Mean Square	*p*-Value
Factor $A_{L}$	214.0261	1	214.0261	0.0308
Factor $C_{L}$	982.8551	1	982.8551	0.0021
Error	80.1261	4	20.0315	
Total	1364.525	8		

$A_{L}$ is the linear component of Adsorbent material factor. $C_{L}$ is the linear component of Mass of adsorbent factor.

**Table 4 materials-13-01824-t004:** Langmuir coefficient comparison with other studies.

Coefficient	Units	Kumar	Habila	This Study
		et al. [27]	et al. [18]	(PVC)
$r^{2}$	Dimensionless	0.996	0.98	0.959
$Q_{0}$	mg/g	21.74	32.11	96.15
*b*	L/mg	0.53	—	0.001
$R_{L}$	Dimensionless	0.54	—	0.944

**Table 5 materials-13-01824-t005:** Mass percent composition of PVC before and after the adsorption process.

Element	Before Adsorption Process % Weight	Before Adsorption Process % Atomic	After Adsorption Process % Weight	After Adsorption Process % Atomic
C	45.6	52.76	47.32	54.56
O	54.36	47.22	52.41	45.37
Al	0.04	0.02	0	0
P	0	0	0.03	0.01
S	0	0	0.05	0.02
Cu	0	0	0.19	0.04
Total	100	100	100	100

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
