# Peer review of "Feasibility Study on the Use of Recycled Polymers for Malathion Adsorption: Isotherms and Kinetic Modeling"

_materials, 2020, doi:10.3390/ma13081824_

Round 1
Reviewer 1 Report
Remarks
Abstract
1). “The adsorption capacity of PVC was three times higher than activated carbon, and a theoretical value of 96.15 mg of Malathion could be adsorbed when using only 1 g of PVC”.
But in conclusion authors writing: “According to equilibrium Langmuir model, a theoretical value of 132.9 mg of malathion could be adsorbed when using only one gram of PVC as adsorbent material”.
2.) “An adsorption velocity rate of 1.98 (1/h) was achieved according to Lagergren kinetic model”.
This value is not adsorption velocity rate but it is the pseudo first-order rate constant (1/h). The adsorption velocity rate has another dimension unit (q/t).
3.) “SEM/EDX analysis, FTIR Spectroscopy and TG/DTA analysis were carried out to demonstrate the adsorption of malathion on these polymers”.
How the results of TG/DTA analysis was demonstrate the adsorption of malathion on PVC? I did not find about it.
Introduction
4.) “The objective of this study is to evaluate the use of these two polymeric materials in the adsorption process of Malathion”
But in this work is mainly present the research data about one polymeric material PVC.
Materials and Methods
5.) “The optimization of the adsorption process was carried out by using a Taguchi L9 orthogonal array”.
This approach should be described in more detail.
6.) “Synthetic waste water with a Malathion concentration of 51.72 mg/L and a pH value of 4.5 was prepared with distilled water”.
Please explain why pH 4.5 was selected.
Results and discussion
7.) “Figure 1a shows that the activated carbon, PVC and HDPS achieved a maximum removal efficiency of 96% when using the maximum volume of dry adsorbent material”.
It is not clear. The adsorption capacity in kinetic study is closed similar for three tested adsorbent but the adsorption capacity in isotherms study (figure 2) is substantially different for all points of experiment for tested adsorbents (activated carbon and PVC).!? It is impossible. Please explain it.
Reviewer 2 Report
Hermosillo-Nevárez et al investigated and compared the adsorption rate of Malathion by PVC, HDPS and activated carbon. The choice of experiments to highlight the findings of the investigation seem logical and well planned. However, there is no new information learned in this study. The results are observational, without any concrete discussion on the differences in the adsorption rates of selected materials. The manuscript at its current state is not up to par for publication.
Specific comments include:
- The particular aims of the study are not well presented in the Introduction, only general descriptions with literatures simply listed.
- The knowledge gap was not clearly identified, authors should rewrite completely this section focusing on the state-of-art on removal of Malathion by adsorbent materials
- Line 20-21: “due to they need large amounts of surface areas for their operation or because of the high costs for their installation and operation”, please rewrite the sentence
- Line 22: “problematic” should be “problem”
- Line 68-69: Polyvinylchloride and Polystyrene have melting point of 100-260oC, how can they be dried under 105oC to eliminate excess water?
- Line 90-92: please provide recovery rate and standard deviation for the extraction method.
- Line 162: In this study, the levels used for temperature were 30oC, 40oC and 50o
- Line 162-163: Authors were planning to use these adsorbent materials in situ in agricultural field, why design the experiment with conditions not correlated with real situation, such as pH 4.5, temperature 40 50 oC, synthetic waste water with a Malathion concentration of 51.72 mg/L and a pH value of 4.5.
- Line 166 and 3.3: section 3.2 and 3.3 are both adsorption kinetics
- Line 173-175: The time required for the removal of organic pollutants depends on adsorbates and reaction conditions, please provide more information for comparison.
- Table 3: the adsorption of Malathion has been investigated by several researchers, it seems like the authors only repeated the previous experiments and compared results, no new methods and findings.
- Authors used three pages to describe results from SEM/EDX analysis, FTIR and TG/DTA, however no solid information was obtained, and those analyses were not mentioned in neither abstract nor conclusions.
Reviewer 3 Report
The present manuscript demonstrates a comparative study of the adsorption capacity of activated carbon, recycled PVC and HDPS for the adsorption of Malathion. The manuscript is well written and studies are informative for the scientific community. I recommend publication of manuscript after minor changes.
- What was the chemical nature and source of waste PVC and HDPS? The adsorption capacity also depends on chemical functional groups on the materials. Is the adsorption was purely physisorption or it was chemisorption.
- The introduction should be modified to show some previous studies on similar topics.
Round 2
Reviewer 1 Report
My advice to the authors for the future, please explain how the study of kinetics and adsorption isotherm are related, especially of the determination of the time needed for equilibrium in adsorption process.
Reviewer 2 Report
The manuscript has been significantly improved, and I do not have any further comments.